



# Near-threshold aeolian sand transport: Effects of boundary layer flow conditions

Ting Jin[1], Lifeng Zhou[2]

[1] School of Metallurgical and Energy Engineering, Kunming University of Science and Technology, Kunming, 650000, China

[2] Yunnan Key Laboratory of Efficient Utilization and Intelligent Control of Agricultural Water Resources, Kunming University of Science and Technology, Kunming, 650000, China

*Corresponding to*: Lifeng Zhou (zhoulf@kust.edu.cn)

**Abstract**

Boundary layer thickness is a critical factor in aeolian sand transport, as it governs the scale of energy-containing turbulent structures, yet its specific mechanisms remain inadequately quantified. Previous studies have established the role of turbulence in particle entrainment but often overlook systematic variations in boundary layer thickness. This study aims to clarify how boundary layer thickness modulates wall-shear stress fluctuations, threshold wind velocities, sand flux, and particle kinematics. We use the three-dimensional large-eddy simulation coupled with a saltation model to investigate these interactions. Results reveal that increased boundary layer thickness enhances extreme-value probability density of wall-shear stress and significantly lowers impact entrainment and rebound thresholds—the latter dropping to less than 50% of conventional wind-tunnel values. Sand transport response is velocity-dependent: at low velocities, transport rises markedly with thickness under fluid-driven entrainment; the effect diminishes at moderate velocities; and at high velocities, transport scales proportionally with thickness under splash-dominated entrainment. Moreover, thicker boundary layers intensify near-bed particle activity, elevating particle velocities and concentrations, reducing intermittency, increasing saltation height, and enlarging mean and variance of airborne particle diameters. These findings elucidate how boundary layer thickness modulates aeolian sand transport via turbulence–particle interactions, offering key insights for improving atmospheric and climate models and advancing the physics of turbulence-driven sediment transport in atmospheric boundary layer.

**Keywords:** Boundary layer thickness; Aeolian sand transport; Turbulent structures; Threshold wind velocity; Atmospheric boundary layer



**1. Introduction**

Wind-driven soil particle movement, also known as aeolian transport, is a key geological and climatic process in arid and desert regions (Shao, 2008). Near-threshold aeolian sand transport occurs around the threshold wind velocity and is characterized by intermittent bursts of intense activity separated by quiescent periods (Stout and Zobeck, 1997; Leenders et al., 2005; Carneiro et al., 2015; Martin and Kok, 2018). Driven by natural wind, this highly unstable process significantly contributes to total mass flux and plays a crucial role in dune evolution, soil erosion, and dust emission. However, its quantitative prediction remains challenging (Martin and Kok, 2018) due to the multiscale nature of turbulent wind fluctuations (Butterfield, 1998; Mathis et al., 2009; Huang et al., 2020; Zhang et al., 2022) and the path-dependent response of sediment transport to these fluctuations (Kok, 2010a).

Accurate prediction of transport rate and intensity is essential for understanding the formation and evolution of aeolian landforms (Sherman et al., 1998). Modeling efforts have combined theoretical, experimental, and numerical approaches. Early theoretical models, such as Kawamura (1951), incorporated a critical shear velocity for particle entrainment (the fluid threshold, $u_*^t = A[gd_p(\rho_p - \rho)/\rho]^{1/2}$, where $d_p$ is particle diameter, $\rho_p$ and $\rho$ are particle and air densities, respectively), and proposed a cubic relationship between transport rate and friction velocity above this threshold, following Bagnold (1941) formulation (coefficient $A = 0.1$). Kok (2010b) later extended White's (1979) model by introducing a probabilistic framework. Wind tunnel experiments have been equally influential: Zhou et al. (2002) tested the Bagnold ($u_* \geq 0.47\ m \cdot s^{-1}$, $u_*$ is friction velocity) and Kawamura ($u_*^t \leq u_* < 0.35\ m \cdot s^{-1}$) equations under different wind velocities and highlighted the central role of threshold velocity. Dong et al. (2003) showed that the threshold coefficient ($A$) decreases linearly with particle Reynolds number. Creyssels et al. (2009) observed a quadratic, rather than cubic, dependence of transport on friction velocity near the threshold, consistent with numerical simulations by Almeida et al. (2006) using Reynolds-averaged methods (critical shear velocity = $0.35\ m \cdot s^{-1}$).

Despite these advances, most models assume steady, continuous sediment transport governed by a single fluid threshold. They fail to capture near-threshold behavior where other critical velocities, such as the impact entrainment threshold (for sustaining continuous transport) and the rebound threshold (for compensating energy loss from particle bouncing), are important. Predictions under such conditions are therefore often inaccurate.

Near-threshold transport is highly intermittent and distinct from steady-state conditions (Rasmussen and Sørensen, 1999). It is strongly influenced by interactions between turbulent coherent structures and sand particles, with different turbulent scales acting through different



mechanisms (Liu et al., 2021). Boundary layer thickness is a key parameter that shapes near-wall
turbulence by influencing the Reynolds number, extent of the logarithmic layer, behavior of large-
scale structures, and distribution of turbulent energy production (Marusic et al., 2017). In wind
tunnels, the boundary layer thickness typically ranges from $0.1 \sim 0.2\ m$ (Clifton et al., 2006;
Parajuli et al., 2016; Li et al., 2020), whereas in the natural atmosphere, it can reach $100 \sim 200\ m$
(Wang and Zheng, 2016). Consequently, even at identical friction velocities, friction Reynolds
numbers may differ by orders of magnitude, leading to marked differences in transport behavior.
Field studies have shown that sediment transport often occurs below the entrainment threshold
in wind tunnels (Rasmussen and Sørensen, 1999), characterized by strong spatiotemporal variability
(Stout and Zobeck, 1997; Baas and Sherman, 2006; Ellis et al., 2012; Huang et al., 2020).
Temporally, intermittent events in the field persist for much longer (Sherman et al., 2013) than in
wind tunnels (Wang et al., 2014). Spatially, transport commonly appears as streamers linked to
large-scale turbulent structures generated higher in the boundary layer (Baas and Sherman, 2005;
Sherman et al., 2013). Streamers in the field can be tens of times longer than those in wind tunnel
experiments (Sherman et al., 2013). Pähtz et al. (2018) emphasized that boundary layer thickness
and turbulent structures are as important as mean shear stress and particle properties in determining
sediment initiation. As a result, conventional incipient motion models—calibrated in wind tunnels—
tend to overestimate the wind velocities required for natural transport. This discrepancy is also
crucial for predicting aeolian activity on extraterrestrial surfaces, such as Mars and Titan, where
boundary layer effects must be considered.
While previous studies have highlighted the importance of turbulent fluctuations, most have
focused on the velocity variability rather than explicitly resolving turbulent structures. For example,
Spies et al. (2000) and Wang and Zheng (2014) introduced periodic velocity fluctuations into steady
winds and observed enhanced transport at low velocities. Kok and Renno (2009) added turbulence
to logarithmic profiles and found that it altered the trajectories of small saltating particles
($d_p < 250\ \mu m$). Huang et al. (2020) further demonstrated the role of unsteady winds in aeolian
transport. However, such studies did not reproduce realistic turbulent structures and capture their
direct influence on particle motion. Dupont et al. (2013) numerically resolved turbulent structures
and reproduced near-surface aeolian streamers, while Wang et al. (2019) showed that streamers form
mainly in the near-wall regions of large-scale structures. More recently, Feng and Wang (2023)
compared transport statistics across boundary layers of different thicknesses, offering insights into
wind tunnel–field discrepancies, though their simulations used friction velocities
($0.43 < u_* < 1.19\ m \cdot s^{-1}$) well above the fluid threshold ($u_*^t = 0.21\ m \cdot s^{-1}$). Jin et al. (2024)
investigated near-threshold transport and identified distinct entrainment mechanisms for rebound



and impact thresholds, showing that particle energy variability influences transport patterns.
Nonetheless, the role of boundary layer thickness in near-threshold aeolian sand transport remains
poorly understood.
To address this gap, the present study builds upon the work of Jin et al. (2024) using three-
dimensional large-eddy simulations coupled with a saltation model. We systematically examine how
boundary layer flow conditions influence both the flow field and near-threshold sediment transport.
Section 2 presents the governing equations, numerical methods, and simulation setup. Section 3
reports the simulation results and analyzes the role of boundary layer thickness. The main findings
are summarized in Section 4.

## 2. Numerical Simulation Approach

The fluid in the boundary layer is assumed incompressible and without thermal exchange. The
dimensionless governing equations are the filtered Navier–Stokes equations:

$$\frac{\partial u_i}{\partial x_i} = 0, \quad \frac{\partial u_i}{\partial t} + u_j \frac{\partial u_i}{\partial x_j} = -\frac{\partial p^*}{\partial x_i} + \nu \frac{\partial^2 u_i}{\partial x_j x_j} + \frac{\partial \tau_{ij}}{\partial x_j} + f_i, \quad (1)$$

where $i = 1, 2, 3$ denote streamwise, vertical, and spanwise directions, respectively, $u_i$ is the
filtered velocity, $t$ is time, $p^*$ is filtered kinematic pressure, $\nu$ is kinematic viscosity, $\tau_{ij}$ is
sub-grid scale (SGS) stress, and $f_i = -1/(\Delta_x \cdot \Delta_y \cdot \Delta_z) \sum_{n=1}^{N_p} f_{Di}$ is the volume force exerted by
particles, where $\Delta_x \cdot \Delta_y \cdot \Delta_z$ is grid volume, $N_p$ is the total number of particles within the grid,
and $f_{Di}$ is the drag force.
Spatial discretization uses a second-order centered finite-difference scheme with a staggered
grid in the vertical direction. Time integration applies a second-order Crank–Nicholson method.
Further implementation details are available in Kim et al. (2002) and Zheng et al. (2020). The
turbulent flow field is initiated by adding random perturbations to the mean laminar wind velocity
profile. Periodic boundary conditions are imposed horizontally, with a stress-free condition at the
top of the domain. At the bottom boundary, the integral wall model proposed by Yang et al. (2015)
is employed due to its superior performance compared to other approaches (Jin et al., 2023). Sub-
grid scale stress is represented using the scale-dependent dynamic model (Porté-Agel et al., 2000),
consistent with Feng and Wang (2023) and Jin et al. (2024).
Particle trajectories are resolved individually in a Lagrangian framework. Particle velocity $u_{pi}$
is given by:

$$m_p \frac{du_{pi}}{dt} = f_{Di} + m_p g \delta_{i2} = \frac{1}{2} C_{dp} A_p \left| u(x_p) - u_p \right| (u_i(x_{pi}) - u_{pi}) + m_p g \delta_{i2} \quad (2)$$

where $m_p$ is particle mass, $C_{dp} = 24(1 + 0.15 Re_p^{0.687}) / Re_p$ is the drag coefficient (Clift et al.,



1978), $A_p = \pi d_p^2 / 4$ is the cross-sectional area of the particle, $Re_p = |u(x_p) - u_p| \cdot d_p / \nu$ is the
particle Reynolds number, and $u(x_p)$ is the fluid velocity at the particle location interpolated with
a third-order Lagrange scheme.

Aerodynamic entrainment is calculated using the residual shear stress rules (Anderson and Haff,

1991; Shao and Li, 1999; Dun and Huang, 2020): $N_a = (m_p \alpha_x u_\tau)^{-1}(\tau - \tau_t)$, where $\tau$ is the local
resolved shear stress, $\tau_t$ is the threshold of aerodynamic entrainment (fluid threshold), $u_\tau$ is the
friction velocity of sand-free flow, and $\alpha_x$ is an empirical coefficient. Liftoff velocity and angle
distributions follow Jin et al. (2024), consistent with the numerical experiments of Jia and Wang
(2021). In addition, a splash function is applied when particles impact the surface, accounting for
both the rebound of incident particles and the ejection of bed particles (Anderson and Haff, 1991;
Dupont et al., 2013). The rebound probability, as well as the velocity and angle distributions of
rebounding particles, together with the number, velocity, and angular distributions of newly ejected
particles, follow the model of Zheng et al. (2020). Bed particles are initially entrained into the
boundary layer by fluid forces, after which the splash mechanism sustains the development of sand
transport. To maintain periodicity, particles exiting the computational domain horizontally are
reintroduced from the opposite boundary, while those escaping from the top boundary are re-
injected into the flow with their vertical velocity reversed.

To examine the effect of boundary layer thickness ($\delta$) on near-threshold transport, two cases

were simulated with $\delta = 5.0$ $m$ and $10.0$ $m$. Results from a smaller domain ($\delta = 1.0$ $m$) partly
draw on Jin et al. (2024). The computational domain dimensions are $8\pi\delta \times \delta \times 2\pi\delta$. Grids are
uniform in the horizontal direction and stretched vertically using a hyperbolic tangent function with
refinement near the wall ($y_1 = 0.012$, $0.014$ $m$ for $\delta = 5.0$ $m$ and $10.0$ $m$). For particle field
post-processing, identical vertical grid resolution was applied to ensure comparability.

Bed particles follow a slightly skewed Gaussian size distribution with a mean diameter of

200 $\mu m$ (Zhu et al., 2019; Liu et al., 2022). Particle and air densities are $\rho_p = 2650$ $kg \cdot m^{-3}$
and $\rho = 1.2$ $kg \cdot m^{-3}$, giving a density ratio of $2208$. The $\alpha_x$ and $\tau_t$ values are consistent
with Jin et al. (2024). Table 1 lists the simulation cases and key parameters.

Table 1. Bulk fluid velocity ($u_b$), saltation friction velocity ($u_*$), Shields number ($\theta_*$
$= u_*^2 / [(\rho_p / \rho - 1)g d_p]$), boundary layer thickness ($\delta$), grid sizes in three directions
($N_x \times N_y \times N_z$), and sand transport rate ($Q$) for 16 simulated cases with sediment transport.

| Cases | $u_b$ ($m \cdot s^{-1}$) | $u_*$ ($m \cdot s^{-1}$) | $\theta_*$ | $\delta$ ($m$) | $N_x \times N_y \times N_z$ | $Q$ ($kg \cdot m^{-1} \cdot s^{-1}$) |
|---|---|---|---|---|---|---|
| 1 | 2.90 | 0.10 | 0.0024 | 5.0 | $512 \times 64 \times 128$ | $4.65 \times 10^{-7}$ |
| 2 | 3.20 | 0.11 | 0.0028 | 5.0 | $512 \times 64 \times 128$ | $4.36 \times 10^{-6}$ |





| 3 | 3.40 | 0.12 | 0.0032 | 5.0 | $512\times64\times128$ | $1.39\times10^{-5}$ |
|---|------|------|--------|-----|------------------------|---------------------|
| 4 | 4.04 | 0.14 | 0.0043 | 5.0 | $512\times64\times128$ | $2.09\times10^{-4}$ |
| 5 | 5.30 | 0.18 | 0.0072 | 5.0 | $512\times64\times128$ | $1.59\times10^{-3}$ |
| 6 | 7.70 | 0.27 | 0.0168 | 5.0 | $512\times64\times128$ | $8.69\times10^{-3}$ |
| 7 | 10.30 | 0.38 | 0.0339 | 5.0 | $512\times64\times128$ | $2.41\times10^{-2}$ |
| 8 | 2.81 | 0.09 | 0.0018 | 10.0 | $768\times64\times192$ | $2.69\times10^{-7}$ |
| 9 | 3.00 | 0.10 | 0.0021 | 10.0 | $768\times64\times192$ | $1.29\times10^{-6}$ |
| 10 | 3.40 | 0.11 | 0.0026 | 10.0 | $768\times64\times192$ | $1.91\times10^{-5}$ |
| 11 | 3.70 | 0.12 | 0.0032 | 10.0 | $768\times64\times192$ | $7.17\times10^{-5}$ |
| 12 | 4.55 | 0.14 | 0.0043 | 10.0 | $768\times64\times192$ | $3.71\times10^{-4}$ |
| 13 | 6.45 | 0.20 | 0.0088 | 10.0 | $768\times64\times192$ | $2.99\times10^{-3}$ |
| 14 | 7.00 | 0.22 | 0.0106 | 10.0 | $768\times64\times192$ | $4.22\times10^{-3}$ |
| 15 | 8.15 | 0.25 | 0.0150 | 10.0 | $768\times64\times192$ | $7.28\times10^{-3}$ |
| 16 | 10.90 | 0.36 | 0.0291 | 10.0 | $768\times64\times192$ | $1.80\times10^{-2}$ |

## 3. Results and Discussion

This section examines how boundary layer thickness influences near-threshold sand transport. The simulations span wind velocities from the rebound threshold up to values exceeding the impact entrainment threshold. To reduce computational cost, each numerically resolved particle represents multiple physical particles (Dupont et al., 2013), with the representative ratio ranging from 50 to 2000 depending on boundary layer thickness and friction velocity. The analysis begins with mean wind velocity profiles and wall-shear stress fluctuations, followed by transport behavior and particle dynamics.

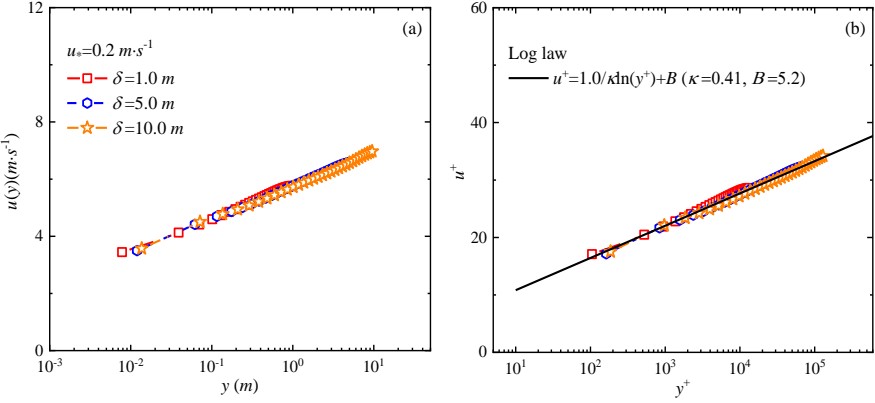

Fig. 1. (a) Simulated mean wind velocity profiles for three boundary layer thicknesses ($\delta = 1.0$, $5.0$, $10.0$ $m$); (b) inner-scale normalized profiles compared with the logarithmic law ($u_\tau = 0.21$ $m\cdot s^{-1}$).

Fig. 1(a) shows mean wind velocity profiles for three boundary layer thicknesses at a fixed





friction velocity, plotted in log-linear coordinates. Profiles overlap closely, with only minor
differences at the first grid point for the smallest boundary layer. Near-wall velocities remain
consistent across all cases, confirming that the first-grid-point height has a negligible influence.
Normalizing the profiles using inner scales ($u^+ = u / u_\tau$, $y^+ = u_\tau y / v$) (Fig. 1(b)) shows excellent
agreement with the logarithmic law, validating the simulated mean flow fields across boundary layer
thicknesses.

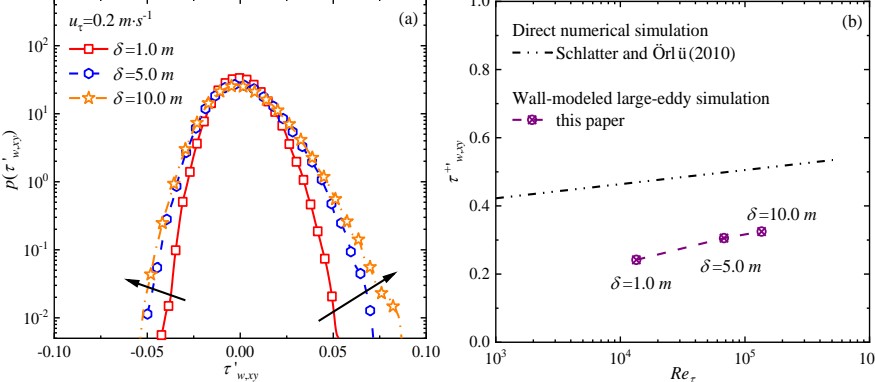


Fig. 2. (a) Probability density distributions and (b) standard deviations of wall-shear stress
fluctuations for different boundary layer thicknesses ($\delta = 1.0$, 5.0, 10.0 $m$).

Particle liftoff is initiated by instantaneous high shear stresses or local pressure imbalances
generated by turbulent fluctuations. Boundary layer thickness influences the velocity threshold for
entrainment by modulating near-wall turbulent structures and the resulting wall-shear stress field
(Lu et al., 2005; Pähtz et al., 2018). Fig. 2(a) shows the probability density distributions of wall-
shear stress fluctuations under the same free-stream wind velocity. The simulations reveal clear
differences across boundary layer thicknesses. As the boundary layer increases, the probability
densities at both tails of the distribution—especially for positive fluctuations above the mean—also
increase. This trend arises because the boundary layer thickness constrains the largest turbulent
scales (Pähtz et al., 2018). A thicker boundary layer supports a broader range of turbulent scales,
producing stronger instantaneous wall-shear stresses. When the boundary layer thickness increases
fivefold (from 1.0 $m$ to 5.0 $m$), the fluctuation amplitude rises markedly, but further increases
lead to a slower rate of growth. Fig. 2(b) compares the standard deviation of wall-shear stress
fluctuations with the direct numerical simulation results of Schlatter and Örlü (2010). The lower
values obtained here reflect the use of wall-modeled large-eddy simulations with relatively coarse
grid resolution. Despite this, the Reynolds number dependence across different boundary layer
thicknesses is well captured.





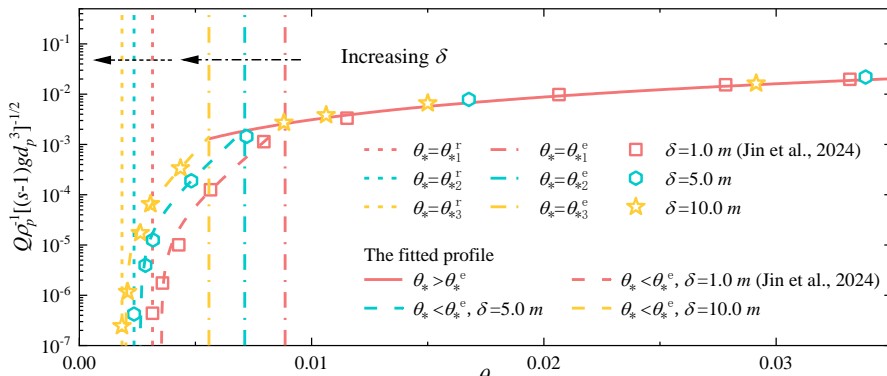

Fig. 3. Simulated sand transport rates under different boundary layer thicknesses

($\delta = 5.0, 10.0 \ m$) and wind velocities.

Large-scale turbulent structures carry significant energy and Reynolds stress (Guala et al., 2006; Balakumar and Adrian, 2007), thereby enhancing energy transfer (Marusic et al., 2010; Serafimovich et al., 2011). The influence of boundary layer thickness on these large structures can further affect particle motion in sand-laden flows. Under simulated conditions with $\delta = 1.0 \ m$, Jin et al. (2024) reported that above the impact entrainment threshold ($\theta_*^e$), the time-averaged sand transport rate scales shear stress raised to the power of 1.5, whereas below $\theta_*^e$, it varies exponentially with shear stress. As shown in Fig. 3, the simulated sand transport rates across different boundary layer thicknesses and dimensionless wind velocities follow the same trend, with fitted curves yielding a high correlation coefficient ($R^2$). However, the threshold wind velocities depend strongly on the boundary layer thickness. For example, the impact entrainment thresholds required for sustained continuous transport are $\theta_{*2}^e = 0.00712$ and $\theta_{*3}^e = 0.00558$ for $\delta = 5.0 \ m$ and $10.0 \ m$, respectively (dot-dashed lines in Fig. 3). These correspond to impact threshold wind velocities ($u_*^e$) of 0.18 and 0.16 $m \cdot s^{-1}$, equal to 0.58 and 0.52 times the fluid threshold ($u_*^t = 0.31 \ m \cdot s^{-1}$). Similarly, rebound thresholds were $\theta_{*2}^r = 0.00235$ and $\theta_{*3}^r = 0.00184$ (dashed lines in Fig. 3), corresponding to rebound threshold wind velocities ($u_*^r$) of 0.1 and 0.09 $m \cdot s^{-1}$, or 0.32 and 0.29 times the fluid threshold.

For a particle size of 200 $\mu m$, the threshold coefficient in a fluctuating flow field is about 1.5 times that in the time-averaged flow (Li et al., 2020a). Based on the entrainment threshold of $u_*^t = 0.21 \ m \cdot s^{-1}$ obtained from wind tunnel experiments, the rebound thresholds are 47.6% and 42.9% of this value, respectively. Field studies also indicate that transport may occur when the friction velocity is just 50% of the wind-tunnel threshold (Rasmussen and Sørensen, 1999). Given measurement uncertainties and the difficulty detecting particles close to the bed (Jin et al., 2021),





the thresholds under field conditions may be even lower than those estimated here. Fig. 4(a) further
shows that both $\theta_*^e$ and $\theta_*^r$ decrease with increasing boundary layer thickness, with the decline
in $\theta_*^e$ more pronounced.

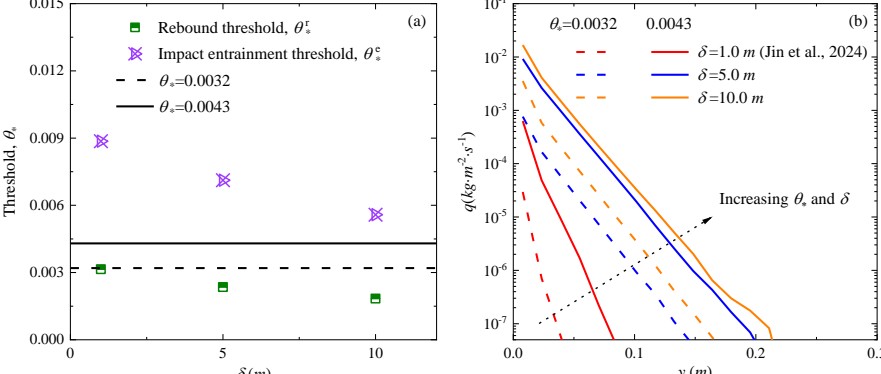

Fig. 4. (a) Rebound and impact entrainment thresholds and (b) sediment transport intensity for

different boundary layer thicknesses (Data for $\delta = 1.0\ m$ taken from Jin et al. (2024)).


Notably, when $\theta_* > \theta_*^e$, the differences in sand transport rates across varying boundary layer

thicknesses become negligible. In contrast, when $\theta_* < \theta_*^e$, the sand transport rate scales with the
boundary layer thickness and rises sharply with incresing wind velocity (Rasmussen and Sørensen,
1999). For example, at $\theta_* = 0.0043$, the transport rates for $\delta = 5.0\ m$ and $10.0\ m$ are 19 and
33 times that for $\delta = 1.0\ m$, respectively; at $\theta_* = 0.0032$, the corresponding factors increase to
29 and 149, demonstrating that the influence of boundary layer thickness is more pronounced at
lower wind velocities. These findings suggest that in real field conditions, sediment transport rates
may be higher and threshold wind velocities lower than predicted in conventional wind tunnels.
Feng and Wang (2023) reported a similar trend, observing that sediment transport rates increase
with boundary layer thickness at wind velocities ($\theta_* > 0.15$, $u_* > 0.8\ m \cdot s^{-1}$ in their study) well
above the near-threshold regime considered in this study. This implies that the effect of boundary
layer thickness on sediment flux depends on the wind velocity and the dominant particle entrainment
mechanism.

Specifically, at wind velocities below the impact entrainment threshold ($\theta_*^e$), thicker boundary

layers generate higher instantaneous wall-shear stresses, enhancing fluid-driven particle flux and
increasing the sand transport rate. When wind velocities far exceed the impact entrainment threshold
($\theta_* > 21\ \theta_{*2}^e$ or $> 27\ \theta_{*3}^e$ according to Feng and Wang (2023)), splash-driven entrainment
dominates, and the sand transport flux becomes approximately proportional to the boundary layer
thickness. In the transitional wind velocity regime between these limits, both fluid- and splash-





driven processes are relatively insensitive to boundary layer thickness, resulting in minimal variation in transport rates.

Sediment transport intensity, which quantifies the non-uniform vertical distribution of sand flux, serves as a key metric linking the microscopic mechanisms of aeolian sand movement—such as particle entrainment and collisions—to macroscopic outcomes, including the overall sediment transport rate. Using the same grid resolution (grid size of $\delta = 1.0 \ m$), Fig. 4(b) shows how sediment transport intensity varies with height for different boundary layer thicknesses ($\delta = 5.0, \ 10.0 \ m$) and dimensionless shear velocities ($\theta_* = 0.0032, \ 0.0043$). For comparison, simulation results for $\delta = 1.0 \ m$ (Jin et al., 2024) are also included to highlight the combined effects of wind velocity and boundary layer thickness. All profiles exhibit an exponential decay with increasing height.

As illustrated in Fig. 4(a), the selected wind velocities ($\theta_* = 0.0032, \ 0.0043$) are above the rebound threshold but below the impact entrainment threshold for all three boundary layer thicknesses, indicating that sediment transport occurs intermittently under these conditions. As both wind velocity and boundary layer thickness increase, the sediment transport intensity rises across all heights, with differences becoming more pronounced at greater heights. The effect of boundary layer thickness is particularly significant at lower wind velocities. For instance, at a height of $y = 0.04 \ m$, the sediment transport intensity for $\delta = 5.0 \ m$ and $10.0 \ m$ increases by approximately 1000 and 3000 times, respectively, relative to $\delta = 1.0 \ m$ at $\theta_* = 0.0032$. At $\theta_* = 0.0043$, the corresponding increases are about 100 and 150 times, indicating that the influence of boundary layer thickness diminishes as wind velocity increases. Importantly, the variations in sediment transport intensity due to boundary layer thickness at this height are far larger than those observed in the total transport rate, since the sediment transport intensity for $\delta = 1.0 \ m$ is relatively low and contributes only minimally to the overall flux.

Fig. 5(a) shows the vertical profile of mean horizontal particle velocity. Unlike continuous transport conditions—where wind velocities exceed the impact entrainment threshold and thicker boundary layers generally result in faster particle movement at the same wind velocity (Feng and Wang, 2023)—the relationship under sub-threshold conditions is non-monotonic. At different wind velocities, particle velocity for $\delta = 1.0 \ m$ is lower than for $\delta = 5.0 \ m$, but shows little change when the boundary layer thickness increases further to $\delta = 10.0 \ m$. As wind velocity rises, the velocity difference between $\delta = 1.0 \ m$ and $\delta = 5.0 \ m$ or $10.0 \ m$ diminishes. Simulation results for $\delta = 5.0 \ m$ and $10.0 \ m$ also demonstrate that near-wall particle velocity is proportional to wind velocity (see inset of Fig. 5(a)), confirming that sediment transport remains intermittent when $\theta_* < 0.0043$ (Jin et al., 2024). However, greater boundary layer thickness



reduces this intermittency, leading to smaller velocity variations across different wind velocities
under thicker boundary layers.

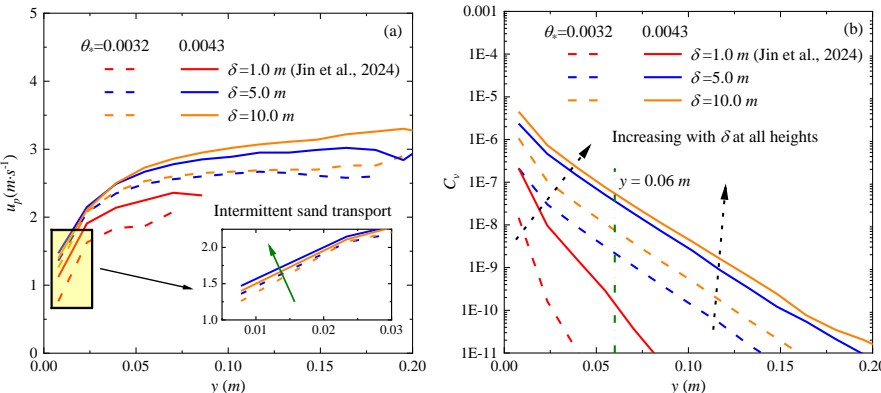


Fig. 5. Vertical profiles of (a) mean particle velocity and (b) particle volume fraction for different
boundary layer thicknesses ($\delta = 5.0$, $10.0$ $m$).

Feng and Wang (2023) observed that particle volume fraction increases with boundary layer

thickness only in regions far from the wall (e.g., $y > 0.06$ $m$ when $\theta_* = 0.0427$). In contrast, the
present results show that under the same wind velocity, particle volume fraction is proportional to
boundary layer thickness across all heights (Fig. 5(b)). This discrepancy arises due to the
predominance of fluid-driven particle entrainment under low wind velocities rather than splash
events. These fluid-driven particles move at lower velocities, and only a small fraction gains
sufficient energy to reach the saltation layer. Consequently, near-wall particle concentration exhibits
a strong dependence on boundary layer thickness. Supporting this, Jin et al. (2024) showed for
$\delta = 1.0$ $m$ that when $\theta_* = 0.0032$ (very close to the rebound threshold), the transport flux is
almost entirely carried by fluid-driven particles. Because such particles have much lower energy
than splash-entrained ones, their flux decays rapidly with height. As wind velocity and boundary
layer thickness increase—where a thicker boundary layer at the same wind velocity corresponds to
a larger argin above the rebound threshold—the decay rate of particle flux with height decreases
progressively.

As wind velocity approaches the rebound threshold, the height of particle saltation decreases.

To illustrate how particle distributions vary with wind velocity and boundary layer thickness, Fig. 6
shows instantaneous particle fields at a representative moment after the aeolian sand flow has
reached a steady state for $\delta = 1.0$ $m$ ($\theta_* = 0.0032$, $0.0043$), $\delta = 5.0$ $m$ ($\theta_* = 0.0032$), and
$\delta = 10.0$ $m$ ($\theta_* = 0.0032$). Particle colors denote velocity, and each plotted particle represents 50
actual particles. For $\delta = 1.0$ $m$ at $\theta_* = 0.0032$, the maximum saltation height is about $0.03$ $m$



(roughly 150 particle diameters), indicating weak sand transport (Fig. 6(a)). Particle motion is
confined to creep or short saltation near the wall, with particle detachment relying primarily on
turbulent fluctuations rather than interparticle collisions. As wind velocity increases ($\theta_* = 0.0043$,
Fig. 6(b)), particle motion intensifies, velocities rise, and the aeolian sand flow develops more
rapidly with increasing boundary layer thickness. Under $\delta = 10.0 \ m$, the maximum saltation height
approaches $0.2 \ m$. Statistical results confirm that at higher wind velocities, increases in flux are
dominated by higher particle concentrations (Fig. 5).

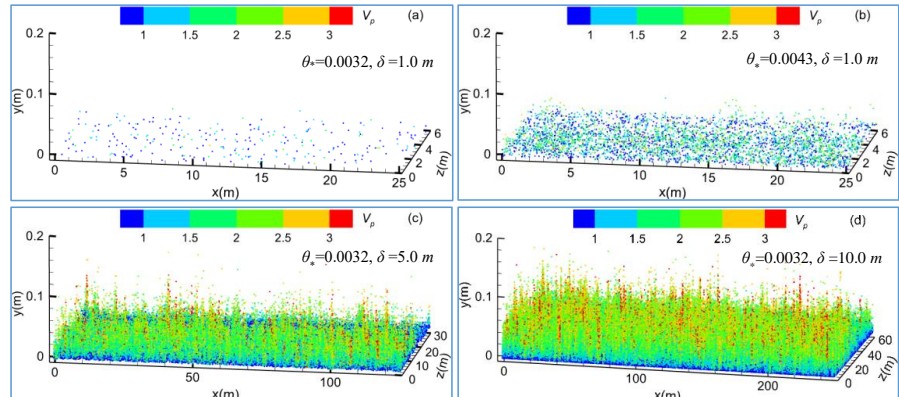


Fig. 6. Instantaneous particle fields for different boundary layer thicknesses

($\delta = 1.0$, 5.0, 10.0 $m$) and wind velocities: (a) $\theta_* = 0.0032$, $\delta = 1.0 \ m$; (b) $\theta_* = 0.0043$,
$\delta = 1.0 \ m$; (c) $\theta_* = 0.0032$, $\delta = 5.0 \ m$; (d) $\theta_* = 0.0032$, $\delta = 10.0 \ m$, where data for
$\delta = 1.0 \ m$ come from Jin et al. (2024).

The saltation layer height $z_m$ was also extracted (Fig. 7(a)), defined as the elevation below

which 99.5% of the total mass flux occurs (Dupont et al., 2013). At wind velocities of $\theta_* = 0.0032$
and 0.0043, the saltation layer thickness for $\delta = 10.0 \ m$ is approximately 3.0 and 2.5 times
greater than for $\delta = 1.0 \ m$, respectively. As wind velocity increases further, the differences among
boundary layer thicknesses diminish, especially for $\delta = 5.0 \ m$ and $\delta = 10.0 \ m$.

To quantify the spatial intermittency of particle distributions, we define the particle spatial

occupancy ($\alpha_p$) as the ratio of grid cells containing particles to the total number of grid cells. Using
the instantaneous particle fields shown in Fig. 6, Fig. 7(b) presents the vertical variation of $\alpha_p$
under different conditions. The results show that $\alpha_p$ decays exponentially with increasing height,
reflecting its close relationship to the vertical distribution of particle volume fraction. Near the wall,
$\alpha_p$ for $\delta = 10.0 \ m$ approaches 1, indicating nearly complete grid-cell occupancy. Under the
same wind velocity, $\alpha_p$ for $\delta = 5.0 \ m$ decreases to $\sim 0.4$, indicating spatial heterogeneity in





particle distribution, while for $\delta = 1.0 \ m$, $\alpha_p$ falls sharply to $0.003$, signifying strong spatial
intermittency with particles confined to localized regions of the flow.

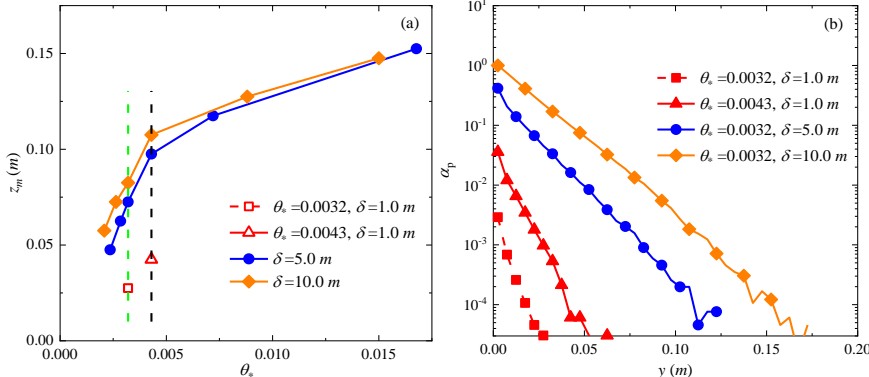

Fig. 7. (a) Saltation layer height and (b) particle spatial occupancy for different boundary layer

thicknesses ($\delta = 5.0, \ 10.0 \ m$) and wind velocities.

Increasing boundary layer thickness markedly enhances energy transfer between the turbulent
flow and the particle phase. Large-scale vortices in thicker boundary layers carry greater energy and
persist longer, which promotes more effective and sustained particle lifting, resulting in both vertical
and horizontal dispersion and thus a more uniform distribution and significantly higher $\alpha_p$ values.
Moreover, the effect of boundary layer thickness on $\alpha_p$ increases with increasing height above the
wall (Fig. 7(b)).

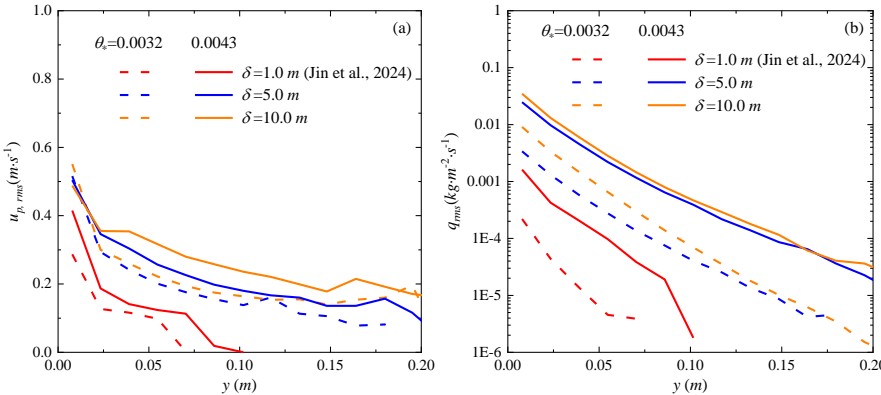

Fig. 8. Vertical profiles of (a) particle velocity and (b) mass flux fluctuations for different

boundary layer thicknesses ($\delta = 5.0, \ 10.0 \ m$).

Fig. 8 presents the vertical profiles of particle velocity and mass flux fluctuations. Even when
the boundary layer thickness increases to $5.0 \ m$ and $10.0 \ m$, the peak of particle velocity



fluctuations remains located in the near-wall region. This near-wall concentration of fluctuations
can markedly intensify wind erosion under low wind velocity conditions. It also reinforces the
prevalence of the intermittent transport regime, dominated by fluid-driven entrainment, which
differs from the continuous saltation dominated by splash-driven entrainment, where the velocity
fluctuation peak typically occurs several centimeters above the bed (Feng and Wang, 2023). Across
all simulated wind velocities, increasing the boundary layer thickness from $1.0\ m$ to $5.0\ m$
significantly amplifies the near-wall velocity fluctuation peak. However, further increases to
$10.0\ m$ produces little additional change, suggesting a gradual transition toward splash-driven
entrainment. Differences in velocity fluctuations associated with boundary layer thickness become
more apparent only at higher elevations above the wall.

Near-threshold sediment transport rate fluctuations also differ from those in continuous

transport. As boundary layer thickness increases, the magnitude of transport rate fluctuations rises
but the incremental effect diminishes, particularly at higher wind velocities. Consequently, the
influence of boundary layer thickness on mass flux fluctuations weakens as wind velocity increases.
This behavior mirrors the response of the mean sediment transport rate, reflecting the fact that as
wind velocity approaches the splash-driven entrainment threshold, both fluid- and splash-driven
processes become less sensitive to variations in boundary layer thickness.

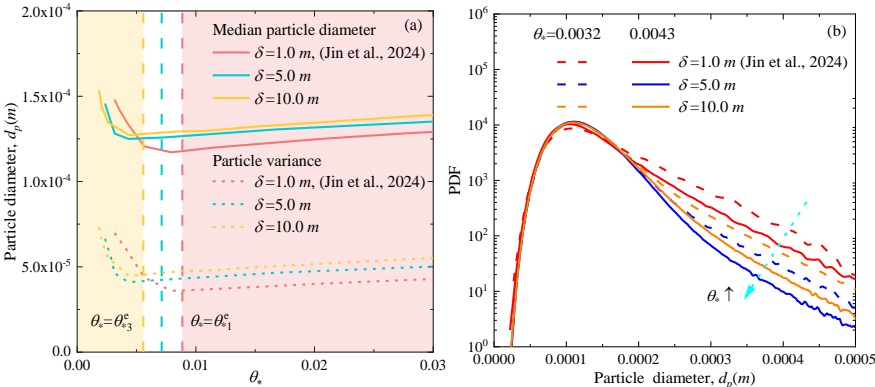


Fig. 9 (a) Mean and variance, and (b) probability density distribution of particle diameter for

different boundary layer thicknesses ($\delta = 5.0,\ 10.0\ m$) as a function of wind velocity.


Under conditions with boundary layer thicknesses $\delta = 5.0,\ 10.0\ m$ (Fig. 9(a)), the variation

of particle diameter parameters reveals two distinct regimes. When wind velocity is below the
impact entrainment threshold ($\theta_*^e$), both the mean and variance of airborne particle diameter
decrease with increasing $\theta_*$. In contrast, once wind velocity exceeds $\theta_*^e$, both parameters become
proportional to $\theta_*$, consistent with the conclusions drawn for $\delta = 1.0\ m$ and supporting the



validity of defining the critical threshold based on transport rate. At lower wind velocities, the
relationship between mean and variance differs across boundary layer thicknesses: $d_{p,ave,\delta=1.0\ m}$
$(/d_{p,var,\delta=1.0\ m}) > d_{p,ave,\delta=5.0\ m} (/d_{p,var,\delta=5.0\ m}) > d_{p,ave,\delta=10.0\ m} (/d_{p,var,\delta=10.0\ m})$ .   Conversely,   thicker
boundary layer thicknesses result in greater mean and variance. Simulation results show that the
critical Shields numbers for $\delta = 10.0\ m > \delta = 5.0\ m$ , and $\delta = 5.0, 10\ m > \delta = 1.0\ m$ are
$\theta_{*1} = 0.003$   and   $\theta_{*2} = 0.005$ , respectively. For wind velocities of $\theta_* = 0.0032$   and   $0.0043$ ,
lying between these two critical values, the relationship between mean and variance shifts
accordingly:   $d_{p,ave,\delta=1.0\ m} (/d_{p,var,\delta=1.0\ m}) > d_{p,ave,\delta=10.0\ m} (/d_{p,var,\delta=10.0\ m}) > d_{p,ave,\delta=5.0\ m} (/d_{p,var,\delta=5.0\ m})$ ,
as also confirmed by the probability density distributions in Fig. 9(b).
As wind velocity increases ($\theta_*$  rising from  $0.0032$  to  $0.0043$ ), the probability of entraining
larger particles decreases because both  $\theta_* = 0.0032$  and  $0.0043$  remain below  $\theta_*^e$ , meaning that
fluid-driven entrainment still dominates particle transport. Under these conditions, the enhanced
near-wall transport flux induces a reduction in local wind velocities due to particle loading (Jin et
al., 2021), which further suppresses the fluid entrainment of larger particles.

**4. Discussion and Conclusions**
This study investigates the role of boundary layer thickness in modulating near-threshold
aeolian sediment transport, a process characterized by high intermittency. Recognizing that
traditional models, often assuming steady, continuous sediment transport governed by a single
threshold (Kawamura, 1951; White, 1979; Creyssels et al., 2009), fail to capture near-threshold
behavior, this research addresses a critical knowledge gap. The primary objective is to
systematically elucidate how different boundary layer conditions influence the turbulent flow field
and the resulting particle entrainment and transport mechanisms near threshold. To achieve this, the
study employs the three-dimensional large-eddy simulation coupled with a Lagrangian saltation
model, aiming to provide a mechanistic understanding of wind tunnel-field discrepancies.
Increasing boundary layer thickness enhances extreme values in wall-shear stress fluctuations.
As a result, both the impact entrainment threshold ($\theta_*^e$  or  $u_*^e$ ) and the rebound threshold ($\theta_*^r$  or
$u_*^r$ ) decrease. For thick boundary layers ($\delta = 5.0\ m$  and  $10.0\ m$ ), the rebound threshold wind
velocity can drop below 50% of values typically observed in conventional wind tunnel experiments.
Sediment transport responds differentially to wind velocity: at very low wind velocities ($\theta_*^r < \theta_*$
$< \theta_*^e$ ), transport increases markedly with thickness under fluid-driven entrainment; at high wind
velocities ($\theta_* > 21\ \theta_{*2}^e$   or   $> 27\ \theta_{*3}^e$ ), it scales proportionally with thickness under splash-driven
entrainment; and at intermediate wind velocities, the effect is negligible. Near-bed particle velocity,
concentration, saltation height, and airborne particle diameter all increase with boundary layer





thickness, accompanied by reduced intermittency and more uniform spatial distributions.
A thicker boundary layer accommodates a broader range of turbulent scales, fostering stronger,
large-scale coherent structures that generate more extreme instantaneous stress events (Pähtz et al.,
2018). This enhanced turbulence facilitates particle entrainment at lower mean wind velocities,
which also explains why the rebound threshold can be less than half the typical wind-tunnel value
(Rasmussen and Sørensen, 1999). Notably, the impact entrainment threshold exhibits a more
pronounced reduction, implying that sustaining continuous transport becomes feasible at relatively
lower velocities as boundary layer thickness increases. Furthermore, the dependence of sand
transport on boundary layer thickness reveals distinct regimes: at low winds, enhanced turbulent
fluctuations directly loft more particles, while at high winds, the system transitions to a splash-
dominated regime where transport capacity scales with the thicker boundary layer (Feng and Wang,

2023).

Thicker boundary layers promote more energetic large-scale turbulent structures that
effectively lift and disperse particles, leading to a more uniform distribution and reduced
intermittency. This mechanism explains previous field observations of longer and more persistent
"streamers" (Baas and Sherman, 2005; Sherman et al., 2013). Unlike the findings of Feng and Wang
(2023), which showed increased concentration only away from the wall, our results reveal the
unique nature of the near-threshold, fluid-entrainment-dominated regime. The observed reversal in
particle size trend is due to the shift from fluid-driven to splash-driven entrainment.
Although this study reveals the significant influence of boundary layer thickness on near-
threshold aeolian sediment transport, several issues require further investigation in the future. This
study examined only three boundary layer thicknesses and a single particle size range. Future work
should extend to thicker boundary layers (closer to realistic atmospheric conditions) and broader
particle size distributions to clarify the underlying mechanisms systematically. The current model
does not account for multiphysical processes, such as interparticle collisions, electrostatic
interactions, or humidity effects, which significantly influence the entrainment and transport of fine
particles in natural environments.
Our findings fundamentally shift how the atmospheric boundary layer should be viewed in dust
emission modeling. By demonstrating that thicker boundary layers can halve the entrainment
thresholds and alter particle size distributions, we provide the mechanistic basis for the known
discrepancy between wind-tunnel models and field observations. This implies that current climate
models likely underestimate dust emissions. Integrating boundary layer thickness into dust emission
schemes is therefore critical for accurate simulation of aerosol radiative forcing, cloud processes,
and the evolution of arid landscapes in a changing climate.




*Data availability*. The data that support the findings of this study are available in the Figshare
repository (https://doi.org/10.6084/m9.figshare.30245776). Additional data related to this paper and
the codes may be requested from the authors.

*Author contributions*. Lifeng Zhou designed and organized the research and its approach. Ting Jin
carried out the simulation, analyzed the results, wrote the manuscript and carefully modified the
manuscript. All authors contributed to the paper.

*Competing interests*. The authors declare that they have no conflict of interest.

*Acknowledgements*. This work was funded by the National Natural Science Foundation of China
(No. 12202170) and the Yunnan Fundamental Research Projects (No. 202301AT070164).

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
