# Peer review of "boundary layer flow conditions"

_EGUsphere, 2025_

## Author Comment (AC1)

This study accurately addresses a long-standing yet insufficiently quantified issue in aeolian physics—the role of boundary layer thickness. Using a large-eddy simulation–saltation coupled model, it systematically reveals how boundary layer thickness modulates turbulent structures and thereby significantly affects the physical mechanisms of near-threshold particle entrainment, transport flux, spatial distribution, and grain-size characteristics. The conclusions provide a clear physical explanation for the discrepancies between wind tunnel and field observations, and offer direct guidance for improving dust emission parameterization schemes in climate models. The paper features a clear structure, sound methodology, and comprehensive data. It is recommended for acceptance after minor revisions. Below are several suggestions for the authors to consider during revision:

Authors' response: Thank you for taking the time to review our manuscript thoroughly and for sharing your insightful comments and valuable affirmation. Based on your suggestions, we have carefully revised and refined the entire manuscript to ensure more concise language and clearer presentation of figures, thereby further enhancing its overall quality. Our detailed point-by-point responses are provided below.

1) To reduce computational costs, the study employs the approach where each numerical particle represents multiple physical particles (lines 166-168), with the representative ratio varying widely (from 50 to 2000) depending on the boundary layer thickness and friction velocity. This is a practical strategy. Please briefly explain the potential impact of this assumption on the key results and its validity, especially under near-threshold conditions characterized by low particle concentration and high representative ratios.

Authors' response: Thank you for your valuable comment. To reduce the computational cost of large-scale particle simulations, this study employed the common approach of representing multiple physical particles with a single numerical particle. We provide below a detailed explanation of the potential implications and the rationale for this assumption.

This methodology primarily influences the precise characterization of particle-particle interactions—notably the splash process—and the statistical robustness under extremely low particle concentrations. Under near-threshold, low-concentration conditions (where the representative ratio is 50:1), the reduced number of numerical particles may introduce slightly greater statistical scatter in the instantaneous particle spatial distribution compared to a fully resolved simulation and could modestly smooth the inherent stochasticity of splash process. However, the central mechanism of this study—that boundary layer thickness modulates near-threshold sand transport by altering large-scale turbulent structures and the resulting extremes in wall-shear stress, thereby governing fluid-driven entrainment—is fundamentally rooted in fluid-particle interactions. This key physics is captured by the accurately resolved flow field and the physics-based drag and entrainment models, which are largely unaffected by the "clustering" of particles in the numerical representation. Consequently, this approach does not compromise core qualitative findings and mechanistic interpretations, such as the "significant lowering of entrainment thresholds" or the "influence of boundary layer thickness on sand transport rate."

This method has been widely adopted in large-eddy simulation studies of wind-blown sand two-phase flow (e.g., Dupont et al., 2013; Feng and Wang, 2023) and has been demonstrated to reliably preserve the accuracy of macroscopic transport statistics. Under near-threshold, low-concentration conditions, splash events are infrequent, and transport is dominated by fluid entrainment. Under these conditions, the effect of particle aggregation on particle-bed interaction statistics is further minimized. Additional sensitivity tests using a lower representative ratio (20:1) confirmed that the influence on our results is negligible. Therefore, within the scope of our research objectives and given practical computational constraints, this approach is both justified and necessary.

[1] Dupont, S., Bergametti, G., Marticorena, B., Simoëns, S., 2013, Modeling saltation intermittency, Journal of Geophysical Research: Atmospheres, 118(13), 7109-7128.

[2] Feng, S. J., Wang, P., 2023, The influences of boundary layer thickness on the characteristics of saltation sand flow–A large eddy simulation study, Aeolian Research, 60, 100853.

2) The friction velocity typically refers to a parameter of the airflow itself, whereas the saltation friction velocity or effective friction velocity often accounts for the feedback from sand particles. Please briefly clarify the specific meaning of the saltation friction velocity used in this paper: is it the bed shear velocity under particle-laden conditions (i.e., the friction velocity that incorporates particle feedback), or is it derived through a specific formulation?

Authors' response: The "saltation friction velocity" used in this study characterizes the actual shear velocity acting on the bed within the fluid–particle two-phase flow system when saltation occurs and reaches dynamic equilibrium. Specifically, at each time step, the model solves the filtered Navier–Stokes equations including the particle drag source term to obtain the realistic flow field. The time-averaged value is then derived from the instantaneous bed shear stress $\tau$ under particle-laden conditions, using the relation $u_*=\mathrm{sqrt}(\tau/\rho)$ (where $\rho$ is the air density). This fully aligns with the definition of an effective friction velocity that accounts for particle feedback. Furthermore, the definition of the saltation friction velocity has been explained in the manuscript.

3) The text mentions classic models such as the cubic law of Bagnold (1941) and the quadratic relationship of Creyssels et al. (2009), and points out that near the threshold state, the relationship between sand transport rate and shear stress follows different patterns (exponential or power law). It is recommended to quantitatively compare the fitted relationships obtained in this study with those from existing research.

Authors' response: Thank you for this important suggestion. We have supplemented the comparison between the fitting relationships obtained in this study and those from classical models.

Specifically, in the "Results and Discussion" section, we clearly state that when the wind velocity exceeds the impact entrainment threshold, the time-averaged sand transport rate obtained from our simulations exhibits a 1.5-power relationship with shear stress. Given that shear stress is proportional to the square of wind velocity, this relationship is equivalent to the sand transport rate being proportional to the cube of wind velocity, which is mathematically consistent with the scaling relationships established by the classical theories of Bagnold (1941) and White (1979) under fully developed, saturated transport conditions. This additional explanation further highlights the consistency between our findings and classical theories in the fully developed transport regime.

At the same time, we have more clearly emphasized the main innovative contribution of this study: it reveals that under near-threshold conditions (where wind velocity is below the impact entrainment threshold), the sand transport rate follows an exponential relationship with shear stress. This fundamentally differs from the continuous, saturated transport patterns assumed by classical models, thereby systematically clarifying the regulatory mechanism of boundary layer thickness in this previously underexplored regime.

[3] Bagnold, R. A., 1941, The physics of blown sand and desert dunes, Springer Netherlands.
[4] White, B. R., 1979, Soil transport by winds on mars, Journal of Geophysical Research, 84(B9), 4643-4651.

4) In the text, the transport intensity is defined as a key metric linking microscopic mechanisms to macroscopic flux, and its variations with height and boundary layer thickness are presented (Fig. 4b). The authors are requested to provide a clear mathematical definition or calculation formula for the transport intensity in the main text (e.g., at line 257), as this would significantly enhance the interpretability and reproducibility of the results in Fig. 4(b).

Authors' response: Thank you for your suggestion. We fully agree that providing a clear mathematical definition for the transport intensity would greatly enhance the readability and reproducibility of Fig. 4(b) and the related analysis.

The transport intensity defined in this study essentially characterizes the non-uniformity of the vertical distribution of sand flux. Specifically, its mathematical expression is the horizontal mass flux per unit height interval. It aims to quantitatively describe the concentration of sand transport activity relative to the total flux at various heights above the bed.

We have supplemented the mathematical expression for transport intensity in the main text as $q(y) = \sum m_p \overline{u_p} / (L_x \times \Delta_y \times L_z)$.

5) Lines 387-389 are slightly cumbersome in syntax. It is suggested to revise them into a clearer structure.

Authors' response: We have revised the relevant text in the manuscript to enhance its clarity and structural logic. The specific modifications are as follows.

The simulation results indicate the existence of two critical Shields numbers: $\theta_{*1} = 0.003$ and $\theta_{*2} = 0.005$. The shift in the particle statistics relationship corresponds to $\theta_{*1}$ when comparing $\delta = 10.0 \ m$ with $\delta = 5.0 \ m$, and to $\theta_{*2}$ when comparing $\delta = 5.0, \ 10 \ m$ with $\delta = 1.0 \ m$.